# Viewing AML through a New Lens: Technological Advances in the Study of Epigenetic Regulation

**DOI:** 10.3390/cancers14235989

**Published:** 2022-12-04

**Authors:** Laura C. Godfrey, Alba Rodriguez-Meira

**Affiliations:** 1Department of Pediatric Oncology, Dana Farber Cancer Institute, Boston Children’s Hospital, Harvard Medical School, Boston, MA 02215, USA; 2Department of Cancer Biology, Dana-Farber Cancer Institute, Boston, MA 02215, USA; 3Broad Institute of MIT and Harvard, Cambridge, MA 02142, USA; 4Department of Haematology, University of Cambridge, Jeffrey Cheah Biomedical Centre, Cambridge Biomedical Campus, Cambridge CB2 0AW, UK

**Keywords:** AML, CRISPR, epigenomic editing, single-cell technology, hematopoiesis, leukemia, histone modifications, DNA methylation

## Abstract

**Simple Summary:**

Epigenetic mechanisms regulate gene expression in each cell type without modifying the underlying genetic sequence. These mechanisms are crucial for normal blood cell function. When they are disrupted, they give rise to diseases such as acute myeloid leukemia, an aggressive type of blood cancer. In this review, we outline the most recent technological advances that enable the study of the epigenetic mechanisms of blood cells with greater precision and a higher resolution. From the technologies that can introduce specific types of epigenetic changes ((epi)-genomic editing) to those that allow us to study the epigenetic mechanisms of individual cells, we summarize the advances which provide a new lens through which to study epigenetic regulation. Given the essential role of epigenetic mechanisms in healthy tissue function and disease, this can provide a comprehensive resource for researchers in the gene regulation field and beyond.

**Abstract:**

Epigenetic modifications, such as histone modifications and DNA methylation, are essential for ensuring the dynamic control of gene regulation in every cell type. These modifications are associated with gene activation or repression, depending on the genomic context and specific type of modification. In both cases, they are deposited and removed by epigenetic modifier proteins. In acute myeloid leukemia (AML), the function of these proteins is perturbed through genetic mutations (i.e., in the DNA methylation machinery) or translocations (i.e., MLL-rearrangements) arising during leukemogenesis. This can lead to an imbalance in the epigenomic landscape, which drives aberrant gene expression patterns. New technological advances, such as CRISPR editing, are now being used to precisely model genetic mutations and chromosomal translocations. In addition, high-precision epigenomic editing using dCas9 or CRISPR base editing are being used to investigate the function of epigenetic mechanisms in gene regulation. To interrogate these mechanisms at higher resolution, advances in single-cell techniques have begun to highlight the heterogeneity of epigenomic landscapes and how these impact on gene expression within different AML populations in individual cells. Combined, these technologies provide a new lens through which to study the role of epigenetic modifications in normal hematopoiesis and how the underlying mechanisms can be hijacked in the context of malignancies such as AML.

## 1. Epigenetic Regulation in the Hematopoietic System

The mammalian genome is finely regulated to ensure precise gene expression in each cell type. This is achieved through epigenetic mechanisms which dynamically control gene expression without perturbing the underlying genetic code. Two well-characterized types of epigenetic marks are histone modifications and DNA methylation. Epigenetic modifier proteins are responsible for either ‘writing’, ‘erasing’, or ‘reading’ these modifications, which can contribute to gene activation or repression.

The epigenetic regulation of gene expression controls differentiation throughout all the stages of hematopoiesis, and its disruption can result in disease. In the context of acute myeloid leukemia (AML), epigenetic dysregulation promotes aberrant gene expression in the hematopoietic stem and progenitor cells, resulting in defective differentiation and leukemic transformation. The disruption of the epigenetic code can occur through the acquisition of genetic mutations in the genes encoding epigenetic regulators or chromosomal translocations that cause epigenetic defects.

In this review, we outline new biological models and technological advances that are instrumental in understanding how histone modifications and DNA methylation regulate transcription in normal cells and how these mechanisms are hijacked in the context of hematologic malignancies such as AML.

### 1.1. Histone Modifications and Their Functions in the Hematopoietic System

Histones can be post-translationally modified with chemical tags such as methylation and acetylation, which are associated with either gene activation or repression. Histone modifications can be used to denote active or repressed cis-regulatory elements, such as enhancers, promoters, and gene bodies [1,2]. Active gene promoters are characterized by both H3K4me3 and H3K27ac, deposited by MLL proteins and p300/CBP, respectively [1,3,4]. Conversely, active enhancer regions are defined by an enrichment of H3K27ac and H3K4me1 (and, in some cases, H3K79me2/3) [1,2,5]. Active gene bodies are demarcated by a non-overlapping pattern of H3K79me2/3 and H3K36me3 at the 5′ and 3′ ends, respectively [6,7,8]. In contrast, H3K27me3, deposited by EZH2, is observed in repressed genomic regions [9]. During hematopoiesis, the balance between repressive and activating histone modifications in both enhancer and promoter regions influence gene expression changes, which can drive differentiation programs.

One histone modification that is particularly important and has been extensively studied for its functional role in the hematopoietic system is H3K79me2/3 (Figure 1a) [10,11]. The H3K79 residue is positioned within the histone octamer core and is methylated in humans by the methyltransferase DOT1L [12]. DOT1L is critical for normal hematopoiesis, with *Dot1l*-deficient mice displaying severe anemia and death at E10.5-13.5 [13]. *Dot1l* deficiency most notably impairs erythroid development, which is coupled with reduced H3K79me levels and the downregulation of the erythroid-specific genes *Gata2* and *Spi1* [14]. DOT1L can mono-, di-, or tri-methylate H3K79 (H3K79me1/2/3), with H3K79me2/3 being the most abundant forms that are highly correlated with gene activity [15]. In a normal cellular context, H3K79me2/3 is linked to transcriptional elongation due to both its position within the 5′ end of the active gene bodies and its ability to directly interact with the components of the super elongation complex (SEC), such as AF9 and ENL (Figure 1a) [16,17,18]. Importantly, AF9 and ENL act as histone acetylation readers which further stabilize DOT1L to chromatin at the gene targets [19]. In addition to transcription elongation, H3K79me is observed at a subset of active enhancers, where it plays an important role in maintaining enhancer–promoter interactions in leukemia models [5,20].

### 1.2. DNA Methylation and Its Function in the Hematopoietic System

DNA methylation takes place through the deposition of a methyl group on the 5′ carbon of the cytosine bases (5mC) at CpG dinucleotides (Figure 1a). DNA methylation primarily regulates gene expression by recruiting or blocking the binding of proteins involved in transcription in the regulatory regions in the genome. DNA methylation is a crucial regulator of normal hematopoietic differentiation, balancing the concerted inactivation of stem cell-associated genes whilst governing the stepwise activation of lineage-defining transcription factors that trigger differentiation [21].

**Figure 1 cancers-14-05989-f001:**
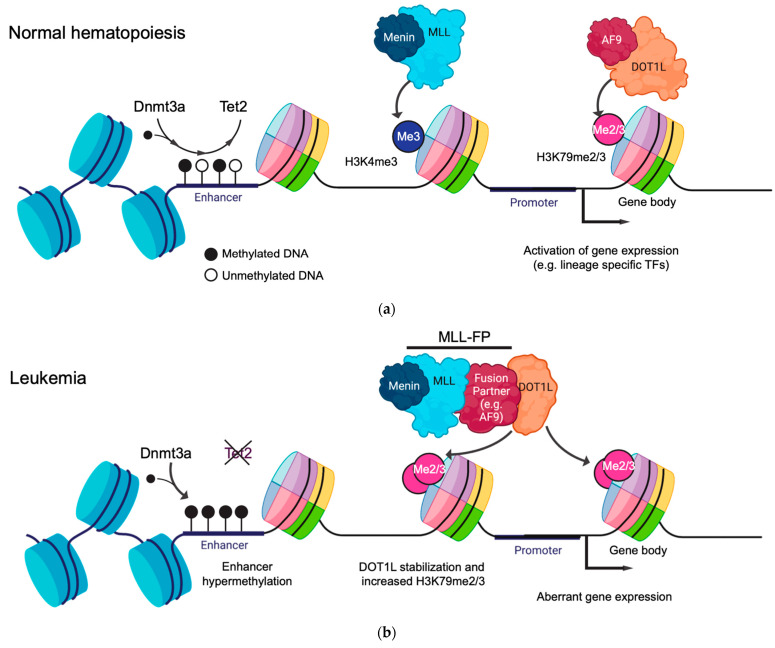
Mechanisms of epigenetic regulation in normal and leukemic hematopoiesis mediated by histone modifications and DNA methylation. (**a**) The de novo DNA methylation machinery dynamically regulates the methylation status of cis-regulatory elements, such as promoters and enhancers, to modulate gene expression. Epigenetic modifier proteins such as MLL or DOT1L catalyze histone methylation in the promoter (H3K4me3) and gene body (H3K79me2/3) regions, respectively, so as to promote gene expression. (**b**) During leukemogenesis, disruption of the DNA methylation machinery, such as *TET2* loss of function, leads to the hypermethylation of enhancers and transcriptional repression. The fusion of epigenetic modifier proteins, such as MLL::AF9, results in the aberrant stabilization of DOT1L and elevated levels of H3K79me2/3, which drives abnormal gene expression signatures.

In the promoter and enhancer regions, DNA methylation inhibits transcription by preventing transcription factor binding and RNA polymerase II activity, resulting in the stable silencing of gene expression [22]. During hematopoietic differentiation, the promoters of lineage-specifying genes and transcription factors are demethylated and transcriptionally activated (such as *POU2AF1*, implicated in B-cell differentiation), whereas the genes required for hematopoietic stem cell self-renewal, such as *MEIS1*, are methylated and silenced [21,23]. DNA methylation also regulates the higher-order 3D chromatin structure by preventing CTCF binding, leading to new topologically associated domain formation, which, in some cases, results in enhancer hijacking and transcriptional activation [24], as in the context of AML [25].

DNA methylation is regulated in mammalian cells by de novo and maintenance pathways, which take place independently of cell division or during DNA replication, respectively. DNA methylation is actively modified in mammalian cells by de novo DNA methyltransferases 3A and 3B (DNMT3A and DNMT3B) and TET enzymes (Figure 1a). DNMT3A/B catalyze the transfer of a methyl group to cytosines, whereas TET proteins catalyze the stepwise oxidation of 5-methylcytosine (5mC) to 5hmC (5-hydroxymethylcytosine) and the subsequent intermediates, which are then converted to unmodified cytosines through the base excision repair (BER) pathway.

DNA methylation is maintained during cell division through the activity of DNA methyltransferase 1 (DNMT1). DNMT1 preferentially recognizes the hemimethylated CpG dinucleotides generated during DNA replication and copies the methylation patterns of the parental strand [26]. Despite this, DNA methylation maintenance is imperfect, especially in genomic regions with a low CpG density [27,28]. This results in a gradual passive loss of DNA methylation over time, a process that is linked to the replicative history and aging of hematopoietic stem cells [29].

### 1.3. Gene Regulatory Function of Histone Modifications and DNA Methylation in Leukemia

Several genetic mutations in histone residues, or in the enzymes which modify them, are observed in AML. Examples of this include mutations in the histone-modifying proteins EZH2, ASXL1, LSD1, or MLL3 [30,31,32,33,34]. Furthermore, H3K27M/I mutant histones predominantly occur in pre-malignant hematopoietic stem cells (HSCs), where they promote self-renewal and leukemogenesis [35,36]. One specific type of leukemia in which a histone modification, H3K79me2/3, has been extensively studied, is MLL-rearranged (MLL-r) leukemia, which is the focus of this review [10]. Many of the cutting-edge technologies that have been used to study H3K79me2/3 and MLL-r AML can be applied to further understand how other histone modifications function in different AML settings.

MLL-r leukemia arises following the chromosomal translocation between the N-terminus of the MLL gene, containing the DNA-binding domain, and the C-terminus of over 100 identified fusion partner genes, with the most common being AF9, AF4, AF10, and ENL [37,38]. This creates an aberrant, functional MLL fusion protein (MLL::FP). Very few co-operating mutations are observed alongside MLL::FPs, indicating that leukemogenesis is driven solely by the MLL::FP itself. MLL::FPs recruit DOT1L to target genes such as the *HOXA* cluster and *MEIS1*. This causes abnormal transcriptional upregulation primarily through the deposition of H3K79me2/3 in their gene bodies and the acquisition of leukemic stem cell properties (Figure 1b) [10,11,39]. The catalytic inhibition of DOT1L leads to the transcriptional downregulation of these target genes and, ultimately, the abrogation of leukemia, indicating that H3K79me plays a critical role in maintaining MLL-r leukemias [40,41,42].

In the clinic, MLL translocations give rise to both AML and acute lymphoid leukemia (ALL), which have a poor prognosis [43,44]. MLL-r leukemia is more common in childhood, as well as those who develop therapy-induced leukemias, particularly those previously treated with topoisomerase II inhibitors [45]. In infants, MLL-r ALL or AML accounts for over 70% of cases acute leukemia [44]. Interestingly, different types of MLL::FPs are more strongly associated with AML, while others are more closely associated with ALL (MLL::AF9 and MLL::AF4, respectively). The mechanisms behind this process are still under investigation, but both the cell of origin and the independent fusion partner function are likely key players. This has been demonstrated in murine knock-in models, in which the *Cre-LoxP* system is used to create in vivo translocations in different cellular contexts. The induction of *Mll::Af9* in primitive progenitor cells using *Lmo2-Cre* gives rise to AML, but not when *Mll::Af9* is induced in other cell types, such as T-cells, or following the induction of different Mll::FPs, such as *Mll::Af4* [46]. This indicates that the cell of origin and the type of MLL::FP expressed are key determinants driving MLL-r AML. Other in vivo models of MLL::AF9 AML rely upon retroviral transduction to overexpress human MLL::AF9 in murine granulocyte and macrophage progenitors (GMP) and Lin-Sca1+kit+ (LSK) cells [47,48,49,50]. The patterns of gene expression and epigenetic landscapes in these models have been shown to faithfully replicate what is observed in MLL-r AML patients. Specifically, elevated levels of H3K79me2/3 have been observed at the MLL::FP gene targets, suggesting that detailed molecular studies based on these models can be used to gain important mechanistic insights into MLL::FP biology [10].

In addition to the major role of histone modification in driving aberrant epigenetic states, DNA methylation is a crucial layer of epigenetic regulation in leukemogenesis. Mutations in *DNMT3A* and *TET2* have been identified in over 60% of individuals with clonal hematopoiesis of indeterminate potential (CHIP) [51,52,53]. Mutations in the DNA methylation machinery have also been found in 44% of patients with AML and many other hematological malignancies, such as myelodysplastic syndromes and myeloproliferative neoplasms [54]. This highlights their role in promoting clonal expansion and leukemogenesis.

Traditionally, the role of DNA methylation in the hematopoietic system has been studied through a series of transgenic mouse models, in which the components of the DNA methylation machinery (e.g., *Dnmt3a*, *Tet2*, *Dnmt1*) are conditionally knocked-out in the hematopoietic system using *Mx1-Cre* or *Vav-Cre*. Conditional *Mx1-Cre-Dnmt3a* and *Vav-Cre-Tet2* knock-out models both displayed increased self-renewal in competitive transplants in vivo, a serial replating capacity, and the aberrant proliferation of the myeloid compartment [55,56]. In *Dnmt3a* knock-out mice, *Gata3* and *Runx1* hypomethylation leads to their overexpression and the concomitant inhibition of hematopoietic differentiation. This provides a mechanistic explanation for the role of DNMT3A in regulating the self-renewal and differentiation of HSCs.

Conditional *Mx1-Cre Dnmt1* knock-out leads to functional defects in self-renewal and increased myeloid differentiation upon competitive transplantation [57,58]. This effect can be attributed to the increased cycling of stem and myeloid progenitor cells, leading to the exhaustion of the stem cell pool. Importantly, *Dnmt1* knock-out or loss of function delays leukemia onset, indicating that functional DNMT1 is required for the self-renewal of both healthy and leukemic stem cells and, therefore, is also implicated in leukemic progression [58,59].

Overall, these studies highlighted the essential roles of histone modifications and DNA methylation in gene expression changes that regulate self-renewal and differentiation in the hematopoietic system and how their disruption leads to clonal expansion and leukemic transformation.

## 2. New Models to Study the Role of Histone and DNA Methylation in AML

Modelling the functional consequences of epigenetic modifications in gene regulation poses many challenges. Firstly, conditional transgenic mouse models do not fully replicate the gene regulatory landscape found in humans, especially in regard to the local CpG or chromatin context regulating specific target genes. Secondly, edited AML cell lines mostly fail to recapitulate the extent of phenotypes associated with the epigenetic mutations commonly found in AML, likely due to already widespread epigenetic changes in the fully transformed stage that AML cell lines are used to model [60]. Recent advances in the ex vivo culturing of mouse and human hematopoietic stem cells, coupled with efficient methods of genome editing, can be used to tackle some of these limitations [61,62,63,64]. These systems provide an unprecedented opportunity to study the effects of epigenetic regulators driving early clonal expansions and eventual leukemic transformation.

The new models used to address these questions in the context of DNA methylation in human cells rely on CRISPR-Cas9 editing using ribonucleoproteins (RNPs) delivered through nucleofection or electroporation. The deletion of *TET2* in cultured umbilical cord blood CD34+ cells, a rich source of HSCs, has been used to model the loss-of-function mutations commonly found in CHIP and myeloid malignancies [65,66]. These models show a competitive advantage of *TET2*-edited cells, with increased myeloid differentiation and reduced 5hmC levels at the promoters and enhancers of genes regulating erythroid differentiation (e.g., leading to a decreased expression of *GATA1* and *KLF1*) (Figure 1b).

New improvements in CRISPR editing now allow for the generation of endogenously induced targeted chromosomal translocations in both in vitro and in vivo settings (Figure 2). This has been used to induce the fusion of the N-terminus of MLL to the C-terminus of either AF4, AF9, or ENL using cultured human CD34+ umbilical cord blood cells [67,68,69,70]. All the in vivo models successfully produced an aggressive acute leukemia. Unlike other models of MLL-r leukemia, which often rely upon the exogenous expression of MLL::FP, the generation of endogenous translocations closely recapitulates the events leading to leukemogenesis in patients. Murine in vitro Mll::Af9 models of MLL-r leukemia have also been generated in murine hematopoietic stem and progenitor cells (HSPCs) using this technology [71]. In addition to human CD34+ cord blood cells, one model of human MLL::AF4 infant ALL was performed by editing fetal HSPC cells [69]. This model gave rise to an aggressive B-ALL which yielded gene expression profiles comparable to those induced in infant MLL-r leukemia patients [69]. On a molecular level, MLL::AF4 gene targets are bound by MLL::AF4 and display elevated levels of H3K79me2 within the gene body. This demonstrates that using CRISPR to generate MLL-r leukemias induces the MLL::FP mechanisms observed in patients and, therefore, can be used to closely recapitulate the disease.

## 3. Epigenomic Editing

Mechanistically dissecting the direct roles of epigenetic modifications in gene regulation is a major challenge for epigenetics research. This requires novel technologies that can be used to perturb epigenetic regulators with precision on a vast scale.

Traditionally, histone modifications in mammalian gene regulation have been closely correlated with active or repressive transcriptional states, but whether they are a direct cause or consequence of such states is still debated. Most proteins catalyzing DNA methylation or histone modifications contain multiple domains and exist in large multi-subunit complexes. Therefore, the complete knock-out of these proteins in order to study epigenetic modifications makes it difficult to detangle catalytic from non-catalytic functions. Indeed, some protein complexes have been shown to play both catalytic and non-catalytic functions in AML, such as the PRC complex [72]. To overcome these limitations, CRISPR-Cas9 has been used to generate deactivating point mutations in the catalytic domains of histone methyltransferases, such as Mll3 and Mll4 (Figure 2) [73,74]. In this example, the impact on transcriptional regulation differed between catalytic inactivation and complete knockout, highlighting fundamental differences between them.

Specifically, the genomic editing of histones is extremely challenging in mammals, primarily due to the abundance of histone-coding genes spanning many different regions of the genome. Recent advances using CRISPR base editing, a technique in which specific point mutations are introduced into the DNA without generating double-stranded breaks, has been used to overcome this limitation in mouse embryonic stem cells (mESCs) [75]. Using an A-to-G base editor, all 28 H3 alleles were edited to generate H3K27R mutant mESCs, resulting in a lack of both H3K27ac and H3K27me3, histone marks associated with transcriptional activity and repression, respectively. Cells lacking H3K27 displayed similar differential gene expression patterns to Suz12 (a PRC2 component which facilitates H3K27me3 deposition) knockout cells. This demonstrates that H3K27me3 plays a direct functional role in gene regulation in mESCs. In contrast, the loss of H3K27ac had no obvious effect on gene expression, RNA polymerase II activity, or the occupancy of the mediator complex at the active enhancers or promoters. This suggests that the function of H3K27ac in maintaining mESC gene regulatory functions is dispensable. In the future, technologies such as CRISPR base editing could be applied to directly investigate the roles of histone modifications in AML.

In addition to the global depletion of histone modifications, CRISPR-based technologies have been repurposed to enable loci-specific epigenomic perturbations. These technologies leverage CRISPR-based systems, in which a catalytically inactive Cas9 protein (“dead Cas9” or dCas9) is fused to an enzyme mediating the addition or removal of epigenomic marks, such as specific histone modifications or DNA methylation (Figure 2) [76]. To deposit specific histone modifications, dCas9 has been fused to the catalytic domains of active chromatin modifiers, such as DOT1L, PRMD9, and p300, or repressive counterparts, such as HDAC3 or the Krüppel-associated box domain (KRAB) [77,78,79]. The recruitment of activating dCas9-p300 to hemoglobin enhancers, usually repressed in HEK293T cells, led to elevated levels of H3K27ac and the activation of transcription [77]. These systems, also known as CRISPRi (CRISPR inhibition) and CRISPRa (CRISPR activation), have been used to perform the genome-wide screening of chemotherapy resistance pathways in AML [80,81].

When dCas9 is fused to an enzyme with DNA methyltransferase activity, this technology enables high-precision CpG editing by using guide RNAs to recruit methylation enzymes to specific sites in the genome. For example, dCas9 fused to DNMT3A or MQ1 (a prokaryotic methyltransferase) were used in primary T lymphocytes and mouse embryos to methylate specific loci in the human genome, such as the RUNX1 gene [82,83]. Upon the targeted methylation of a nearby CTCF site, RUNX1, which is important for leukemogenesis, showed increased expression, likely due to the disruption of the TAD boundaries. Alternatively, TET enzymes fused to dCas9 (dCas9-Tet1) were used to demethylate specific CpG sites and activate the expression of silenced genes [83]. New variants of these systems have been developed using the SunTag system. This system consists of a repeated array of peptide epitopes that recruit multiple copies of antibody-fused DNA methyltransferases to a single locus. This enables higher levels of methylation over broader regions of the genome, such as the entire promoters [84].

A major limitation of the current dCas9-based epigenomic editors is the size of dCas9 (encoded by over 4 kilobases) and its fusion proteins of interest. This makes it difficult to deliver such constructs to hematopoietic cells using retroviral or lentiviral vectors due to limitations on efficient viral packaging. To avoid this limitation, the co-transduction of two lentiviral vectors encoding dCas9-p300 and an sgRNA fused to two different fluorescent proteins was used in mouse primary T-cells [85]. However, co-transduction is not likely to be a widely applicable transgene delivery strategy for all the hematopoietic cell types.

Other strategies that may be used to address this size limitation include the development of new transgenic lines for epigenomic editing: Rosa26:LSL-dCas9-p300 for gene activation and Rosa26:LSL-dCas9-KRAB for gene repression [86]. These mouse lines require the delivery of small sgRNAs to their target regions in the genome, which can be accomplished using conventional nucleofection or transduction methods and would greatly facilitate epigenomic editing on a larger scale.

## 4. Single-Cell Epigenomic Technologies in Normal and Leukemic Hematopoiesis

Genome-wide epigenomic approaches, such as chromatin immunoprecipitation (ChIP-seq), assays for transposase-accessible chromatin (ATAC-seq), and bisulfite sequencing (BS-seq), have been instrumental for the interrogation of histone modification, DNA accessibility, and methylation patterns in bulk populations, respectively. Clear disadvantages of these technologies include the requirement of a large numbers of cells and, importantly, that fact that they mask epigenetic heterogeneity between cells within a population. Some of these technologies have been further developed and adapted for low cell numbers and single-cell inputs, making significant contributions to our understanding of the hematopoietic system and hematopoietic malignancies such as AML.

One of the pioneering single-cell epigenomic technologies is single-cell ATAC-seq (scATAC-seq), now established in widely used droplet microfluidic platforms [87,88]. In fact, scATAC-seq led to the characterization of essential regulatory elements and transcription factors governing lineage decisions in hematopoiesis and demonstrates how these are disrupted in AML [89,90].

The study of histone modifications by ChIP-seq in cases of low cell numbers poses clear challenges related to background signal and cellular crosslinking requirements, although pioneering studies have adapted this technology to droplet microfluidic platforms so as to achieve single-cell readouts [91,92,93]. Newer methods, such as CUT&RUN and CUT&Tag, are much more sensitive, do not require cell fixation, and are more suited for low cell input [94,95]. They rely on Protein A-tagged nucleases (MNase in CUT&RUN and Tn5 in CUT&Tag), which bind antibodies recognizing specific histone modifications, or chromatin proteins, resulting in the cleavage of nearby DNA. High-throughput sequencing of these DNA fragments enables the mapping of histone-bound genomic regions.

Recent advances in single-cell CUT&RUN and CUT&Tag provide a higher-resolution view of the histone modification landscape of individual cells (Figure 2) [96,97]. In the context of MLL-r leukemia, single-cell CUT&Tag has highlighted the presence of both the active (H3K4me3) and repressive (H3K27me3) chromatin states of MLL::FPs gene targets, which differ between individual cells within a cellular population [98]. For example, cells with a high *HOXA9* and *TAPT1* expression displayed high H3K4me3 and low H3K27me3 levels at their promoters. Inversely, low H3K4me3 and high H3K27me3 were associated with low *HOXA9* and *TAPT1* expression. This bivalency could reflect the MLL::FP binding dynamics at its gene targets in individual cells within a leukemic population.

CRISPR tiling is another technology used to explore chromatin modifier functions by coupling single-cell gene expression data with CRISPR editing. This approach has been used to examine the importance of individual DOT1L domains in MLL-r leukemia [99]. In this study, the authors tiled sgRNAs across DOT1L exonic regions, measuring the gene expression and barcodes in single mouse *Mll::Af9* leukemia cells. The sgRNAs targeting the N-terminus of DOT1L, which contains the methyltransferase domain responsible for H3K79me deposition, showed deregulation of DOT1L-inhibitor-sensitive genes, such as *Meis1*, *Hoxa9*, and *Myc*, as well as myeloid differentiation genes, including *Cd11b* and *Gr1*. Single-cell techniques coupled with gene editing approaches can be used to distinguish precisely which individual epigenetic modifier domains are important drivers of leukemogenesis.

Technological developments have also enabled the profiling of DNA methylation in concert with other genomic readouts at the single-cell resolution (Figure 2) [100]. Pioneering work used single-cell multi-omic technologies that capture transcriptomes, mutations, and methylomes from the same single cell to assess the role of DNA methylation in the evolution of chronic lymphocytic leukemia and the effects of *DNMT3A* mutations on clonal hematopoiesis [101,102]. In the latter, they identified the selective hypomethylation of PRC2 targets in *DNMT3A* mutant cells. The hypomethylation of the CpGs flanking PRC2 targets is enriched in binding motifs of hematopoietic transcription factors such as MYC, a key driver of leukemogenesis. The same hypomethylated regions could also be identified in AML samples, suggesting that DNA methylation alterations acquired in the preleukemic stage might have functional consequences during leukemia progression.

In summary, the development of single-cell technologies to study epigenetic regulation in healthy and malignant hematopoiesis has provided critical insights into the cellular hierarchies that construct each state. The wider adoption of these technologies by the hematopoietic community will hopefully lead to a high-resolution picture of the epigenetic mechanisms driving leukemogenesis.

## 5. Discussion

Molecular tools that can be used to interrogate the functional role of epigenetic modifications are rapidly being developed. These technologies, including CRISPR-Cas9 editing and single-cell approaches, represent a new era of research, aiming to study epigenomics from a new perspective at a high resolution (Figure 2). Applying these methods to understand how epigenetic mechanisms are hijacked during leukemogenesis will ultimately guide the development of novel therapies in order to treat hematological malignancies and uncover the basic biological mechanisms that drive them.

CRISPR-Cas9 technologies have been shown to be instrumental for the investigation of AML disease models and our understanding of the underlying epigenetic mechanisms which drive them. Despite their wide application, thus far, there are questions remaining in the MLL-r AML field that CRISPR approaches could be used to explore further. Even though the same fusion proteins, such as MLL::AF9, can be responsible for leukemogenesis, chromosomal translocations can occur at different breakpoints, meaning that not all MLL::AF9 fusion proteins are identical [103,104]. As both MLL and AF9 contain different protein interaction domains, it is unclear whether the inclusion or exclusion of certain MLL::AF9 regions leads to differences in the disease phenotype. To understand this in the future, precise CRISPR-Cas9 editing could be used to generate different endogenous MLL::AF9 translocations.

Another variant of CRISPR-Cas9 editing which has not yet been widely applied to study the role of epigenetic modifications in AML is CRISPR base editing. Given that MLL-r AML is dependent on H3K79me, base editing could be used to create H3K79 mutant MLL::AF9 cells in order to directly investigate how H3K79me impacts transcriptional regulation in this setting. Base editing could be similarly used to study the function of mutant histones, such as H3K27M, in AML settings, as well as wildtype H3K27me3, which is potentially perturbed in AML models carrying *EZH2* mutations. This technology could also be particularly useful as a means to model mutations in the proteins of the DNA methylation machinery. Many of these mutations are hotspot (DNMT3A-R882) or gain-of-function mutations (IDH1-R132 and IDH2-R140). Therefore, base editing could be used to specifically introduce these mutations in an endogenous context without completely knocking out or overexpressing the mutant versions of these genes, more faithfully replicating the context in which they occur in leukemia, as recently shown in *IDH* mutant TF1 cells [105].

The next generation of CRISPR-based tools is epigenomic editors. Overall, epigenomic CRISPR-Cas9 editors are a promising system to perform functional perturbations of a cell’s epigenetic state. However, some of their current limitations include the potential off-target effects of delivering endogenous enzymes (such de novo Dnmts) into the nucleus in large quantities. Studies addressing this issue have found non-specific, widespread methylation independent of sgRNA expression in multiple cell types, indicating that the further refinement of these tools will be necessary to achieve truly targeted epigenomic editing [106]. Furthermore, dCas9, especially when fused to an epigenetic modifier, is extremely large (several hundred kilodaltons). Therefore, when targeted to specific genomic loci, it can interfere with transcription by blocking DNA accessibility and preventing protein binding. Finally, the delivery of large Cas9-containing vectors into cells is still inefficient. In the future, the development of recombinant epigenomic editors, which could be delivered as RNPs, could lead to the higher efficiency of these tools with respect to hematopoietic cells using nucleofection.

In addition to CRISPR-Cas9-based editing, other technologies, such as targeted protein degradation, offer a promising method for studying the functions of histone modifications and, potentially, DNA methylation in AML. Proteolysis-targeting chimeric (PROTAC) technology enables the rapid degradation of endogenously tagged proteins. Recently, this was used in the context of MLL-r leukemias to rapidly degrade MLL::AF9 [42]. Using this system, a pronounced effect on transcriptional elongation was observed after only fifteen minutes of MLL::AF9 degradation. Importantly, PROTAC therapy could be used in the future to rapidly degrade disease-causing epigenetic proteins driving AML.

CRISPR-Cas9 or PROTAC technologies perturb epigenetic protein function by genetic deletion or rapid protein degradation, respectively. In contrast to these approaches, another way to specifically perturb histone modifications and DNA methylation without genetic manipulation is the use of small molecule inhibitors. These are important tools due to the relative ease of their delivery into patients. Indeed, some epigenetic inhibitors are currently being used in the clinic to treat AML patients.

The classical hypomethylating agents used in AML treatment (decitabine and azacytidine) are cytidine analogs irreversibly incorporated into the DNA during replication, which target DNMTs for proteasomal degradation [107]. However, they currently have a limited clinical application due to their lack of efficacy as single-therapy agents, which highlights the need for a new generation of DNMT inhibitors. GSK3685032 is a new selective, non-covalent, and reversible DNMT1 inhibitor which blocks DNMT1 from binding to hemimethylated CpGs generated during DNA replication. This molecule induces DNA hypomethylation, activation of gene expression, and inhibition of leukemia cell growth without the cytotoxic effects associated with nucleoside analogs, such as decitabine. Importantly, it has demonstrated a high efficacy in AML xenograft models, presenting an exciting opportunity for clinical implementation and promise as a valuable tool that could be used to understand the mechanism of action of DNMT1 during AML progression [108].

DOT1L inhibitors (Pinometostat/EPZ-5676), along with SNDX-5613 and KO-539 (targeting MENIN, a protein involved in MLL::FP recruitment), are being used in clinical trials on MLL-r AML, as well as other subtypes, including *NPM1*-mutated AML [38,109,110,111]. These inhibitors lead to the downregulation of MLL::FP target gene expression, alterations in the epigenetic landscape, and the inhibition of leukemic cell growth. Despite showing clinical efficacy, drug resistance and relapse have been reported in the case of both DOT1L and Menin inhibitors [112,113,114]. Even though these are targeted therapies, it is likely that refractory epigenetic mechanisms will arise in single cells. Furthermore, other, more general chemotherapeutic agents, such as anthracyclines, have shown epigenetic-dependent resistance mechanisms in leukemic stem cells [115]. These mechanisms might be overlooked when performing bulk epigenetic and gene expression analyses, such as ChIP-seq or RNA-seq. Single-cell technologies could potentially be used to identify resistant clones and help to develop alternative therapies so as to overcome resistance mechanisms in the clinic. Specially, new technological advances that couple single-cell multi-omic readouts with lineage tracing would allow us to understand how these epigenetic marks are dynamically established as cells divide and differentiate, how they might be clonally propagated to drive AML transformation, and how they are specifically selected to cause resistance.

Despite the fact that both histone modifications and DNA methylation play important roles in gene regulation in AML, the functional interplay between these two types of epigenetic marks remains relatively under-studied. An example of this interplay is the DNA hypermethylation of bivalent domains at the promoters of genes implicated in AML [116]. Another example is the overexpression of DOT1L in *Dnmt3a*-/- murine HSCs [117]. DOTL1 overexpression was coupled with an increase in H3K79me at genomic loci that are prone to loss of DNA methylation and linked with the genes overexpressed in leukemia. The treatment of human *DNMT3A* mutant AML cell lines, xenograft models, and leukemic stem cells with DOT1L inhibitors led to reduced proliferation, cell cycle arrest, and differentiation [117,118]. This suggests that mutations in the DNA methylation machinery can prime cells to become dependent on specific histone modifications. Further insights are required to elucidate this interplay in AML and to understand how imbalances in DNA methylation patterns can influence histone modification deposition and vice versa.

## 6. Conclusions

Histone modifications and DNA methylation play integral roles in maintaining normal hematopoietic development, and their disruption leads to aberrant gene expression patterns which ultimately drive disease. New technological advances such as (epi)genomic editing and single-cell methods will continue to unlock important insights into how epigenetic modifications impact gene regulation in healthy tissues and in the context of disease.

## Figures and Tables

**Figure 2 cancers-14-05989-f002:**
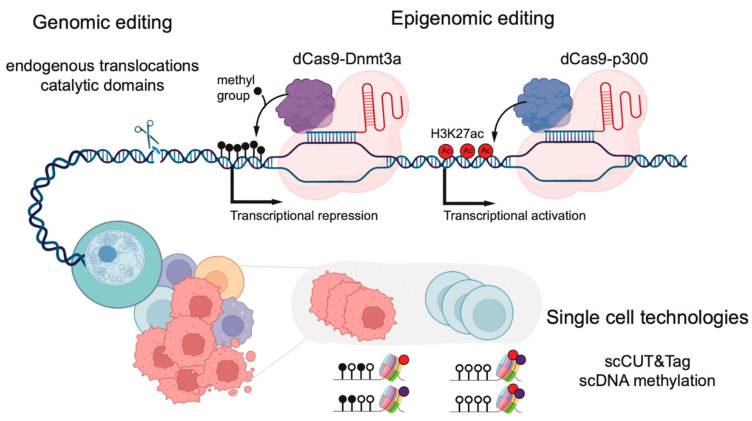
New technologies used to study epigenetic regulation in the hematopoietic system. dCas9 fused to proteins catalyzing DNA methylation (e.g., DNMT3A) or depositing histone modifications (p300; H3K27ac) can be used to modulate epigenetic marks in targeted genomic regions. Newly developed CRISPR-Cas9 systems can also be used to generate endogenous translocations between epigenetic regulators, and single-cell multi-omic technologies provide novel insights into the layers of epigenomic regulation in individual cells.

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
