# Peer review of "Viewing AML through a New Lens: Technological Advances in the Study of Epigenetic Regulation"

_cancers, 2022, doi:10.3390/cancers14235989_

Round 1

Reviewer 1 Report

In this work, Godfrey LC and colleagues reviewed epigenetic modifications in hematopoietic system and technological advances to study epigenetic dysregulation in AML.

The review included relevant figures and a constructed text.

I find this work interesting. I can come with some hopefully constructive comments:

Comments :

1/ The authors could include and discuss the results presented in this study:

Van den Boom V et al. Non-canonical PRC1.1 Targets Active Genes Independent of H3K27me3 and Is Essential for Leukemogenesis. Cell Reports. 2016

2/ Regarding single cell DNA methylation analyses, the authors could present and discuss this manuscript:

Karemaker ID et al. Single-Cell DNA Methylation Profiling: Technologies and Biological Applications. Trends Biotechnol. 2018

3/ The authors could include and discuss these results related to 3D genome organization in AML:

Xu J et al. Subtype-specific 3D genome alteration in acute myeloid leukaemia. Nature 2022.

4/ The authors could discuss the results presented in : Van Gils N et al. Targeting histone methylation to reprogram the transcriptional state that drives survival of drug-tolerant myeloid leukemia persisters. IScience 2022.

5/ What is known about epigenetic dysregulation and leukemic stem cells?

Reviewer 2 Report

It is a very good review article ! Definitely accepted for pubblication. The authors might pay attention to the Scalea S. et al work and find the way to cite this article (Scalea S. et al The FEBS Journal 2020).

Reviewer 3 Report

In this review,the authors outline the most recent technological advances that 1enable the study of epigenetic mechanisms in blood cells with greater precision and at higher resolution. Firstly, Histone modifications and their function in the hematopoietic system, especially DNA methylation in leukemia ,were introduced. Moreover, the mechanism on new models to study the role of histone and DNA methylation in AML was elucidated, as well as single cell epigenomic technologies in normal and leukemic hematopoiesis. This review summarizes the cutting-edge research technology for epigenetic dysregulation in AML. However, the manuscript requires minor revisions.

1.Please define all abbreviations in the text when used for the first time. For example, in page 2 ,line 74 DOT1L.
